# Interest in long-acting injectable ART among adolescents with perinatally acquired HIV in Cameroon: Implications for implementation in developing countries

Yagai Bouba[1,2]*, Aude Christelle Ka'e[1], Cynthia Ayafor[1], Lum Forgwei[1], Jeremiah Efakika Gabisa[1], Daniel Mabongo[3], Alex Durand Nka[1], Rita Ekwoge Mejane[4], Suzie Ndiang Tetang[5], Rachel Simo Kamgaing[1], Francis Ndongo Ateba[6], Nadine Nguendjoung Fainguem[1], Michel Carlos Tommo Tchouaket[1], Desire Takou[1], Nelly Kamgaing[1], Michelle Aloum Menye[7], Félicité Noukayo[8], Ezechiel Ngoufack Jagni Semengue[1], Roland Wome Basseck[1], Agabus Wiadamong[1], Abdou Rahamani Gnambi[1], Catherine Eyenga[5], Naomi-Karell Etame[1,9], Aurelie Minelle Kengni Ngueko[1,9], Larissa Gaëlle Moko Fotso[1,10], Junie Flore Yimga[1], Grace Anong Beloumou[1,9], Collins Ambe Chenwi[1,9], Samuel Martin Sosso[1], Alice Ketchaji[11], Hyppolite Kuekou Tchidjou[12], Gregory Edie Halle Ekane[13], Paul Ndombo Koki[6], Daniele Armenia[2], Vittorio Colizzi[14], Rogers Awoh Ajeh[13,15], Gianluca Russo[4], Stefano D'amelio[4], Francesca Ceccherini-Silberstein[9], Carlo-Federico Perno[16], Alexis Ndjolo[1,10], Maria Mercedes Santoro[9], Joseph Fokam[1,3,10,13]

1 Chantal Biya International Reference Centre for Research on HIV/AIDS Prevention and Management (CIRCB), Yaoundé, Cameroon, 2 Faculty of Medicine, Saint Camillus International University of Health Sciences, Rome, Italy, 3 Central Technical Group, National AIDS Control Committee (NACC), Yaoundé, Cameroon, 4 Department of Public Health and Infectious Diseases, University of Sapienza, Rome, Italy, 5 Essos Hospital Centre, Yaoundé, Cameroon, 6 Mother and Child Centre, Chantal BIYA Foundation, Yaoundé, Cameroon, 7 Mbalmayo District Hospital, Mbalmayo, Cameroon, 8 Cité Verte District Hospital, Yaoundé, Cameroon, 9 Department of Experimental Medicine, University of Rome Tor Vergata, Rome, Italy, 10 Faculty of Medicine and Biomedical Sciences, University of Yaoundé I, Yaoundé, Cameroon, 11 Department of Disease, Epidemics and Pandemic Control, Ministry of Public Health, Yaoundé, Cameroon, 12 Paediatrics Emergency Services, Amiens University Hospital, Amiens, France, 13 Faculty of Health sciences, University of Buea, Buea, Cameroon, 14 UNESCO Board of Multidisciplinary Biotechnology, University of Rome "Tor Vergata", Rome, Italy, 15 Global fund and partners Grant Coordination Unit for AIDS, Tuberculosis and Malaria, Ministry of Public Health, Yaoundé, Cameroon, 16 Bambino Gesù Children Hospital, IRCCS, Rome, Italy

* romeobouba@yahoo.fr

## Abstract

Long-acting injectable antiretroviral therapy (LAI-ART) has the potential to transform HIV treatment for adolescents with perinatal HIV infection (APHI). Given the existing knowledge gaps surrounding this recent strategy, we aimed to assess interest in LAI-ART among ART-experienced APHI in Cameroon. We conducted a cross-sectional study among ART-experienced APHI aged 10–19 years, across four paediatric HIV clinics in the Centre region of Cameroon. Data were collected from 4 to 30 November 2024 using structured questionnaires to assess sociodemographic characteristics, awareness of, and interest in LAI-ART. HIV-1 viral load (VL) and CD4 cell counts were measured during the same period. Logistic regression analysis was used to

**Data availability statement:** All relevant data underlying the findings of this study are included in the manuscript and its Supporting information files (S1 Dataset and S1 Data Dictionary). The dataset has been de-identified to protect participant confidentiality.

**Funding:** This study was funded by the International AIDS Society (IAS) through the Collaborative Initiative for Pediatric HIV Education and Research (CIPHER): CIPHER-ADOLA-2023 (CIPHER Grant ID 0323 to YB). The funders had no role in study design, data collection and analysis, decision to publish, or preparation of the manuscript.

**Competing interests:** The authors have declared that no competing interests exist.

identify factors associated with interest in switching to LAI-ART. Of 248 participants (48.8% female; median [IQR] age 15 [13–17]), 40.0% lived with a guardian and 75.8% had partial/full HIV-disclosure; 3.2% reported stigma. Clinically, 77.5% were in multi-month ARV-dispensation (MDSD) and 28.2% had poor ART-adherence; fear of injection was none/moderate/high in 73.4%, 15.7% and 10.9%, respectively. Viral undetectability (VL < 50 copies/mL) and CD4 ≥ 500 cells/mm³ were 71.6% and 72.5%, respectively. Overall, 92.3% expressed interest in LAI-ART, motivated by reduced pill burden/fatigue (63.7%) and expected better adherence (74.2%); 30.2% reported prior LAI-ART awareness. In multivariable analysis, no prior awareness had 81% lower odds of interest (p = 0.036). Compared with no-fear, moderate-fear was associated with 46% lower odds (p = 0.393), while high-fear was associated with 78% lower odds (p = 0.017). Disparities in sex, education, residence, adherence, virological response, or treatment history did not influence interest in LAI-ART. Among all respondents, 77.1% said they would choose LAI-ART if available, whereas 20.2% preferred MDSD. In this Cameroonian APHI cohort, awareness of LAI-ART was low despite high-interest once adolescents were informed. Rollout in low/middle-income countries should prioritise targeted education, youth-friendly counselling to close knowledge gaps, reduce fear and ensure equitable uptake.

## Introduction

According to the World Health Organization (WHO), 39.9 million people live with HIV, 80% of the 140000 adolescents affected reside in sub-Saharan Africa (SSA) [1]. In Cameroon, approximately 490484 people were living with HIV in 2023, including more than 18000 adolescents aged 15–19 years [2]. Interestingly, adolescents living with HIV (ADLHIV) experience poor health outcomes in the HIV care cascade [2]. The availability of second-generation integrase strand-transfer inhibitors (INSTI)-based antiretroviral treatment (ART), administered orally, has significantly improved the clinical outcomes of PLHIV at the global level, including ADLHIV [3]. Despite progress with oral ART regimens that offer good tolerability and fewer side effects, single tablet regimens (fixed dose combinations) and once daily pills, there is persistent adherence challenges among ADLHIV. This calls for innovative therapeutic approaches to optimise ART outcomes of ADLHIV to achieve the 95% UNAIDS goal of viral suppression (VS) toward HIV elimination by 2030 [4]. Of note, recent data from Cameroon showed a low VS rate of 74.4% in ADLHIV as compared to adults (90.8%) [5]. This shortfall might be driven by pubertal changes that alter drug pharmacokinetics, emerging HIV 1 drug resistance (HIVDR) and, most importantly, inconsistent adherence to antiretroviral therapy [6,7]. Inconsistent adherence may reflect limited access to comprehensive services tailored for adolescents. This is further compounded by developmental factors such as growing autonomy, peer influence and continuing cognitive maturation that erode the discipline required for daily dosing [6,7]. To mitigate adherence-related challenges, strategies such as support groups,

differentiated service delivery (DSD) models and counselling have been implemented to help adolescents cope with the aspects of living with HIV, which could be substantially improved by strategies to reduce the burden of daily medication. Existing adherence supports for adolescents such as peer groups, DSD and tailored counselling address psychosocial barriers but still depend on daily pill taking and leave a key obstacle unresolved. Among the various strategies, long-acting injectables (LAI), a novel delivery system involving monthly or bimonthly injections, show considerable potential to address some of these challenges and support sustained VS among ADLHIV [8].

The Food and Drug Administration has recently approved a LAI-ART (LAI-ART), notably cabotegravir (CAB) which is a 2nd generation INSTI and rilpivirine (RPV) which is a 2nd generation non-nucleoside reverse transcriptase inhibitor (NNRTI), as an alternative to daily pills to maintain VS [9,10]. The implementation of LAI-ART has shown promising results, particularly in improving adherence and VS [10,11]. Adherence to ART was reportedly improved when compared to daily oral treatment. Participants credited LAI-ART not only as having a superior VS capacity but also as being a redemption from the daily psychological reminder of living with HIV, enhancing privacy in HIV care and treatment, and reducing HIV-related stigma associated with taking oral pills amongst the others [12]. In low- and middle-income countries (LMICs), studies increasingly show that LAI-ART has a high potential acceptability, especially among those facing adherence issues associated with stigma, discrimination, or violence [13,14]. According to a study in Uganda, it was observed that, after switching to LAI-ART, the majority of PLHIV requested to remain on this novel therapeutic regimen [13]. Experts also suggest that modest adherence gains during the formative 12–19-year window may translate into decades of viral suppression and faster progress toward the 95-95-95 targets, while warning that equitable rollout is imperative so that rural and low-income youth are not left behind [15]. Despite the fact that not all adolescents can benefit from this new therapeutic strategy (eligibility criteria), there is a tremendous enthusiasm and hope among ADLHIV [16] but evidence for a successful implementation and scale-up in LMICs remains limited and even substandard [15,17–19]. Interest in LAI- ART has been also growing in high income countries, particularly as an alternative to daily oral medication [20]. To address barriers to uptake, current implementation strategies aim to enhance accessibility and promote patient engagement [21]. Consequently, LAI-ART may represent a transformative advancement in ART, with the potential to sustain optimal VS in LMICs [22].

Although CAB + RPV LA is not yet available in most LMICs, ongoing advocacy seeks to expand this therapeutic option for ADLHIV. Moreover, in LMICs, a large proportion of ADLHIV are perinatally infected, representing a particularly vulnerable subgroup that faces unique clinical and psychosocial challenges, including pill fatigue, stigma, mental health burdens, and difficulties in maintaining long-term adherence [23]. Despite the promise of LAI-ART in addressing these barriers, there remains a substantial lack of evidence on its operational feasibility, particularly the health system preparedness, service delivery strategies, and adolescent preferences. This gap is particularly evident in Central African countries like Cameroon, where no quantitative data exist on adolescent interest or acceptability of LAI-ART. Furthermore, it remains unclear how APHI perceive LAI-ART relative to existing DSD approaches such as multi-month dispensing, which shows promises and is increasingly deployed in sub-Saharan Africa (SSA) [24]. To address these evidence gaps, the "ADOdolescent and Long-Acting antiretroviral treatment (ADOLA)" Study is conducted within the framework of the Collaborative Initiative on Paediatric HIV Education and Research (CIPHER) [25–27]. As part of global efforts to support the introduction and scale up LAI-ART among APHI, we sought to evaluate the level of knowledge/awareness and interest to LAI-ART as well as its predictors among APHI in the Cameroonian context.

## Methods

### Ethics statement

This study adhered to all the principles outlined in the Helsinki Declaration for conducting research on human subjects. Specifically, authorisations were obtained from each participating sites (from the 4 clinical sites and CIRCB); ethical clearance was obtained from the Centre Region's Ethics Committee under reference CE N°0056 CRERSHC/2023 on 14

March 2023 (initial approval) and CE N°0892 CRERSHC/2024 on 22 August 2024 (extension approval). An administrative approval was obtained from the regional delegation of public health. A written parental consent was obtained from legal parents/guardians and all participating APHI 12–19 years provided their written informed assent prior to enrolment. Prior to enrollment, the study's information form was shared to all legal parents/guardians and all participating APHI 12–19 years, and concerns were addressed accordingly. Data confidentiality was ensured by using unique identifiers while anonymity was ensured by avoiding any disclosure of participants' status as per guidelines.

### Study design

A cross-sectional study was conducted among APHI (10–19 years) receiving Tenofovir/lamivudine/dolutegravir (TLD) within the CIPHER-ADOLA cohort in the centre region of Cameroon. Briefly, CIPHER-ADOLA is a project aimed at characterising the clinical and genotypic profiles of ART-experienced APHI, their eligibility and willingness to receiving LAI-ART in a near future [26].

To capture a more comprehensive understanding of APHI knowledge and interest in LAI-ART in Cameroon, we adopted a heterogenous multi-site study, involving four HIV treatment clinical centres in the Centre region, one located in rural setting (Mbalmayo district hospital) and three in urban setting (city of Yaoundé: Mother-Child Centre of the Chantal Biya Foundation, the Essos Hospital Centre, the Cité Verte).

### Sample-size determination

Sample size was estimated by using the formula $n = Z^2\, p(1-p)/d^2$, applying a 95% confidence level (Z = 1.96), an expected LAI-ART acceptability of p = 0.65 [14], and a desired absolute precision of d = 0.06. This yielded 243 participants, which was inflated by 5% for potential non-response to give a final target of 256 participants. We enrolled 248 adolescents (97% of target), providing an achieved half-width of ±5.9%, essentially identical to the planned ±6%.

### Eligibility criteria and enrolment of participants

Following a consecutive sampling, participants were enrolled from the CIPHER-ADOLA cohort based on the following inclusion criteria: (a) APHI aged 10–19 years, (b) initiated or switched to the TDF/3TC/DTG regimen, (c) provided a written parental consent and a written assent from the adolescents 12 years above. Non-inclusion was based on incomplete relevant data for the study and the inability to provide the required blood sample for testing. The sampling process was handled by a study team composed of epidemiologists, paediatricians, nurses, virologists, psychosocial counsellors and the representatives of the community of young people living with HIV.

### Data collection and definition of variables

Data were collected from 4 to 30 November 2024 through face-to-face interviews using a structured questionnaire. Socio-demographic (age, sex, area of residence, stigma, drug use, educational level), HIV history (HIV disclosure status, ART regimens, ART adherence, ART duration), knowledge and interest in LAI-ART were collected as continuous or categorical variables wherever appropriate. Knowledge of LAI-ART (yes/no) was defined as having heard of its use for HIV treatment, regardless of depth of understanding. Interest in LAI-ART (yes/no) was defined as willingness to switch to this treatment if locally available. The expressed interest in LAI-ART represents a hypothetical acceptability based on a general preference after participants were informed about an injectable ART option administered every 8 weeks at the follow-up unit. However, this measure reflects a stated preference rather than a firm intention to switch, particularly given the limited depth of knowledge regarding potential side effects, logistical requirements, and long-term implications. ART adherence was assessed over the past 30 days based on Cameroon national guidelines and categorized as good (≤1 missed dose, ≥95%), moderate (2–4 missed doses, 85–94%), or poor (≥5 missed doses, <85%) [28]. Body mass index (BMI) was

assessed using sex-specific BMI-for-age percentiles and classified as underweight (<5th percentile), normal (5th to <85th percentile), or overweight (85th to <95th percentile) [29]. Immunocompromised status was defined as a CD4 count <500 cells/μL, further classified as moderate (200–500 cells/μL) or severe (<200 cells/μL) [30]. VS was defined as the measurement of plasma VL < 1000 HIV-RNA copies/mL in accordance with the WHO programmatic threshold [31]. Viral undetectability was defined as a plasma viral load <50 HIV-RNA copies/mL, reflecting the clinical viral suppression target typically used to determine eligibility for LAI-ART. Fear of injections or needles was assessed using a closed-ended question with three response categories: no fear, moderate fear, and high fear. Interest in LAI ART (yes/no) among APHI was defined as the primary outcome variable, while all other variables were treated as independent predictors.

### Laboratory tests

Blood samples used for the laboratory analyses were collected on the same day the interview was performed, and all the laboratory tests were carried out at the Chantal Biya International Reference Centre (CIRCB). Briefly, CD4 T-lymphocyte count was performed on whole blood using the CyFlow Counter flow cytometer (Sysmex Corporation, Kobe, Japan), following the manufacturer's instructions; and HIV-1 RNA viral load quantification was performed on plasma samples using the Abbott m2000rt real-time PCR platform (Abbott Molecular, Des Plaines, IL, USA), following the manufacturer's instructions.

### Statistical analyses

All collected data were entered into a Microsoft Access database with restricted access and the quality control of data was ensured by double checking of data entry by a second investigator and then validated by the supervisor. Statistical analyses were performed using SPSS version 26. Categorical variables were compared using Chi-square or Fisher's test as appropriate. A binary logistic regression analysis was performed to identify the factors associated to the interest of APHI to LAI-ART. Multivariate logistic regression analysis was performed to ascertain predictors of interest to LAI-ART by including all the variables with p < 0.25 in the univariate model [32]. Model adequacy was appraised after fitting the final logistic regression: calibration was tested with the Hosmer–Lemeshow goodness-of-fit statistic (10 risk deciles), and multicollinearity was gauged by variance-inflation factors (VIFs). We regarded a Hosmer–Lemeshow p > 0.05 and VIFs ≤ ≈10 as acceptable; any borderline excess was scrutinised but retained when substantively justified. The threshold for statistical significance was set at p < 0.05.

## Results

### Characteristics of the study population

Overall, 248 eligible APHI were enrolled (median [IQR] age 15 [13–17] years and 48.8% females) and 75.8% had a complete/partial HIV status disclosure. Only 3.2% and 5.6% reported being stigmatised and oral-drug-users respectively. Concerning ART adherence, poor/moderate adherence in the last 30 days was observed in 28.2% of the individuals. Regarding ART modality, 77.5% were in a multi-month ARV dispensing differentiated service delivery (MDSD) model, 84.3% had achieved VS defined as VL < 1000 copies/mL (Table 1), while 71.4% had an undetectable VL (VL < 50 copies/mL). Concerning the immunological status, the median (IQR) CD4 count was 674 (470–910) cells/mm$^3$, with 71.8% having a CD4 count above 500 cells/mm$^3$.

### Interest of APHI in LAI-ART

Only 30.2% of the APHI were aware of the existence of LAI- ART, a remarkable proportion (92.3% [95% CI 88.3% – 95.0%]) showed an interest in LAI-ART versus only 7.7% (n = 19) showing no interest, and both males and females showed a similar high interest in LAI- ART, p = 0.192, similar with other stratified variables (Table 1). The distribution of

**Table 1. Characteristics of APHI with perinatal infection, overall and stratified according to their interest in LAI-ART.**

| Variable | Count, N = 248 | Interested in LAI-ART | | p-value |
|---|---|---|---|---|
| | | No 19 (7.7%) | Yes 229 (92.3%) | |
| Sex, female, n (%) | 121 (48.8) | 12 (9.9) | 109 (90.1) | 0.192 |
| Age groups, n (%) | | | | |
| 10-14 | 119 (48.0) | 9 (7.6) | 110 (92.4) | 0.955 |
| 15-19 | 129 (52.0) | 10 (7.8) | 119 (92.2) | |
| Health facility settings, urban, n (%) | 209 (84.3) | 17 (8.1) | 192 (91.9) | 0.517 |
| Education level, n (%) | | | | |
| No formal Education | 12 (4.8) | 1 (8.3) | 11 (91.7) | 0.934 |
| Primary | 47 (19.0) | 3 (6.4) | 44 (93.6) | |
| Secondary/University | 189 (76.2) | 15 (7.9) | 174 (92.1) | |
| Living with biological parents, yes, n (%) | 150 (60.5) | 8 (5.3) | 142 (94.7) | 0.088 |
| Body mass index group, n (%) | | | | |
| Underweight | 21 (8.5) | 0 (0.0) | 21 (100.0) | 0.383 |
| Normal weight | 205 (82.7) | 17 (8.3) | 188 (91.7) | |
| Overweight | 22 (8.9) | 2 (9.1) | 20 (90.9) | |
| HIV disclosure status, n (%) | | | | |
| Not disclosed | 60 (24.2) | 7 (11.7) | 53 (88.3) | 0.180 |
| Complete or Partial disclosure | 188 (75.8) | 12 (6.4) | 176 (93.6) | |
| Age at disclosure, median (IQR)* | | | | |
| <10 years | 21 (8.5) | 2 (9.5) | 19 (90.5) | 0.346 |
| 10–19 years | 167 (67.3) | 10 (6.0) | 157 (94.0) | |
| ART adherence in the last 30 days, n (%) | | | | |
| Poor | 17 (6.8) | 0 (0.0) | 17 (100.0) | 0.179 |
| Moderate | 53 (21.4) | 2 (3.8) | 51 (96.2) | |
| Good | 178 (71.8) | 17 (9.6) | 161 (90.4) | |
| Non-injecting drug use, yes, n (%) | 14 (5.6) | 1 (7.1) | 13 (92.9) | 0.940 |
| Experienced stigma in the past 12 months, yes, n (%) | 8 (3.2) | 1 (12.5) | 7 (87.5) | 0.601 |
| CD4 cell count (cells/mm³, median (IQR) | 674 (470-910) | 644 (531-911) | 674 (462-910) | 0.416 |
| Viral load levels (copies/mL), n (%) | | | | |
| <50 | 177 (71.4) | 11 (6.2) | 166 (93.8) | 0.376 |
| 50- 999 | 32 (12.9) | 4 (12.5) | 28 (87.5) | |
| >1000 | 39 (15.7) | 4 (10.3) | 35 (89.7) | |
| Knowledge of LAI-ART, n (%) | | | | |
| Yes | 75 (30.2) | 2 (2.7) | 73 (97.3) | 0.052 |
| No | 173 (69.8) | 17 (9.8) | 156 (90.2) | |

**Note:** *Calculation was based on adolescents with partial or complete HIV disclosure (n = 188). ART: antiretroviral therapy; LAI-ART: long-acting injectable ART; IQR: interquartile range.

CD4 cell counts showed that only 2% of participants had <200 cells/mm³, while the majority (71.8%) had CD4 counts between 200 and 500 cells/mm³. Among those interested in LAI-ART, 80% were in the <200 cells/mm³ group, compared to 98.5% and 90.4% in the 200–500 cells/mm³ and >500 cells/mm³ groups, respectively (p = 0.132).

Regarding the fear of injection/needle by APHI, 73.4%, 15.7% and 10.9% had no, moderate- and high fear of injection/needle, respectively. When stratifying the fear of injection according to the interest of APHI to LAI-ART, a significant

difference was found in favour of those with no fear (Fig 1A). Of note, a higher proportion of APHI with high injection fear was found in those not interested in LAI-ART compared to those interested (31.6% versus 9.2%, p = 0.010).

Among APHI who expressed interest in LAI-ART, the distribution of viral load categories did not differ significantly across fear levels (p = 0.810), suggesting a broad acceptability of this modality irrespective of viral load levels status (Fig 1B).

Several reasons justified the interest of APHI to LAI-ART, the majority of APHI expressed that it will facilitate ART adherence (74.2%) and relives from pill fatigue (63.7%) (Fig 2).

Finally, we have also collected data on the preference of APHI between LAI-ART and MDSD model for at least 3 months, showing that APHI largely preferred LAI-ART (77.1%) over MDSD model (20.2%), p < 0.001; but 2.8% were undecided (p < 0.001).

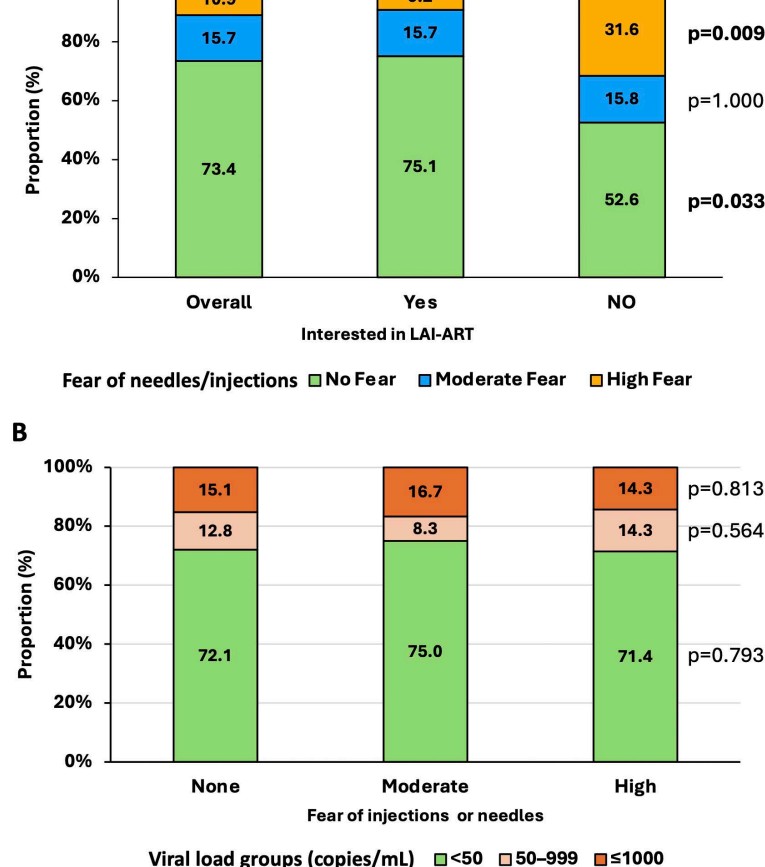

**Fig 1. Interest in LAI-ART by injection fear level and viral load status among adolescents with perinatally acquired HIV infection.** Panel A shows that most adolescents reported no fear of injections, with broadly similar distributions across levels of interest in LAI-ART, although higher fear appeared more frequent among those not interested. Panel B suggests comparable distributions of viral load categories across injection fear levels, with no clear pattern of association.

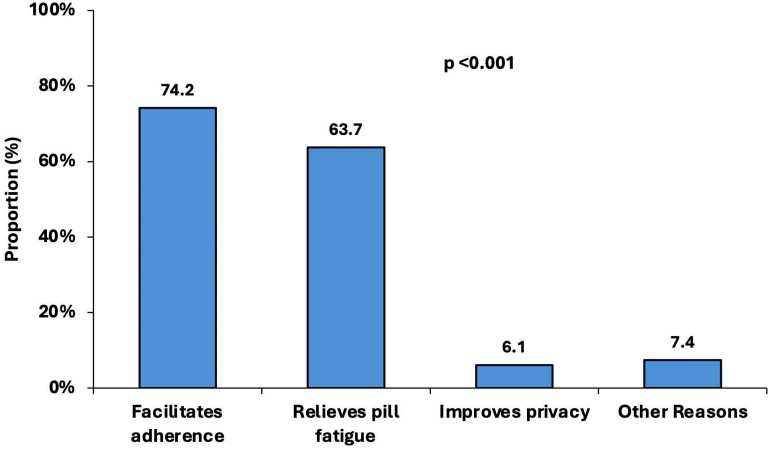

**Fig 2. Drivers of interest in long-acting injectable ART among adolescents with perinatally acquired HIV.** Interest in LAI-ART was primarily driven by perceived adherence benefits and relief from pill fatigue, with fewer adolescents citing privacy or other reasons. Note: multiple responses were allowed.

## Determinants of Interest in LAI-ART among APHI

Based on variable selection criteria for the multivariate logistic regression analysis, sex, living with biological parents, HIV status disclosure, knowledge of LAI-ART, CD4 cell count, and fear of injection were included in the multivariate model. After adjusting for these variables, it was observed that knowledge/awareness on injectable ARVs and the fear of injection/needle were the only predictors of interest in LAI-ART among APHI (see Table 2). Of note, APHI reporting moderate fear of injection had 46% lower odds of interest in LAI-ART compared with those reporting no fear (aOR 0.54, 95% CI 0.13-2.23, p = 0.393), while those with high fear had 78% lower odds (aOR 0.22, 95% CI 0.06-0.76, p = 0.017). Moreover, APHI lacking prior awareness of LAI-ART were 81% less likely to express interest compared with those who knew about it (adjusted OR 0.186, 95% CI 0.038- 0.900; p = 0.036).

## Discussion

Despite the availability of modern ART such as TLD, an optimal adherence is required to achieve and maintain the control of HIV replication. This is particularly true for APHI who continue to struggle with compliance to ART (with the daily oral intake), thus raising potential hope with the advent of LAI-ART and especially for such vulnerable population [13,15,33]. Today, LAI-ART is not available in most developing countries, however, over 90% of Cameroonian APHI expressed interest in LAI-ART, despite only one-third having heard of it, regardless of being male or female.

In this population of APHI wherein about 40% appeared to be orphans, the social and psychological vulnerability regarding daily ART compliance would be more detrimental [34]. Even though 8 out of 10 APHI have reached the secondary level of education (in accordance with the median age of 15), the finding might differ from the national context since participants (84.3%) were in treatment centres located in urban settings. With few cases of stigmatization and non-injecting drug use reported in this study, interest to LAI-ART might be minimally impacted by these conditions. Importantly, with up to 24% of undisclosed HIV status, adherence and interest to LAI-ART might be substantially affected. Of note, WHO recommends that children of school age should be told of their HIV-positive status; younger children should be informed incrementally in preparation for full disclosure. HIV disclosure is an important process in the care of APHI, owing to its role on ART adherence and treatment outcomes [35]. In fact, our data show that only about 72% APHI did not report any missed ART dose in

**Table 2. Predictors of interest in long-acting injectable ART among adolescents with perinatal HIV infection.**

| Variable | Univariate analysis | | Multivariate analysis | |
|---|---|---|---|---|
| | OR (95% CI) | p-value | aOR (95% CI) | p-value |
| **Sex** | | | | |
| Male | 1 | | 1 | |
| Female | 0.530 (0.201–1.394) | 0.198 | 0.367 (0.128–1.057) | 0.063 |
| **Living with biological parents** | | | | |
| Parental | 1 | | 1 | |
| Guardian | 0.446 (0.172–1.151) | 0.095 | 0.606 (0.213–1.728) | 0.349 |
| **HIV status disclosure** | | | | |
| Not Disclosed | 1 | | 1 | |
| Full/partial Disclosure | 1.937 (0.726–5.169) | 0.187 | 1.448 (0.481–4.363) | 0.511 |
| **Knowledge of Injectable LAI-ART** | | | | |
| No | 0.251 (0.057–1.117) | 0.070 | 0.186 (0.038–0.900) | **0.036** |
| Yes | 1 | | 1 | |
| **CD4 cell count, cells/mm³** | | | | |
| <200 | 1 | | 1 | |
| 200-500 | 8.500 (0.441–163.885) | 0.156 | 19.554 (0.843–453.383) | 0.064 |
| >500 | 2.368 (0.250–22.409) | 0.452 | 3.219 (0.288–35.985) | 0.343 |
| **Fear of injection/needle** | | | | |
| No fear | 1 | | 1 | |
| Moderate fear | 0.698 (0.183–2.663) | 0.598 | 0.539 (0.130–2.229) | 0.393 |
| High fear | 0.203 (0.067–0.617) | 0.005 | 0.218 (0.063–0.758) | **0.017** |

Note: Age groups, education level, body mass index, viral load levels, age at HIV disclosure, experienced stigma, Non-injecting drug use, ART adherence and living with biological parents had a $p < 0.25$ in the univariate model. Model validation: Hosmer-Lemeshow goodness-of-fit $\chi^2(8) = 11.0$, $p = 0.200$; maximum variance-inflation factor (VIF) = 10.5 for the CD4 category dummies, with all other VIFs ≤ 1.1—indicating acceptable calibration and no problematic multicollinearity.

the past 30 days. It is therefore crucial to mitigate poor adherence by implementing interventions with high potential such as the LAI-ART that could improve health outcomes and reduce AIDS-related mortality among APHI.

The low rate of knowledge of LAI-ART among APHI may affect real-world uptake, emphasizing the need for greater awareness campaign in preparation of the introduction or transition to this new treatment option, taking into consideration that 9 out of 10 APHI may have interests.

High acceptability of LAI-ART has been consistently reported across diverse settings, with studies in Europe indicating interest rates above 65% among people living with HIV [36]. Evidence from high-income countries, including IMPAACT 2017 and HPTN 084–01 [8,18] further supports strong adolescent acceptance. Our findings align with these trends but highlight additional barriers in LMICs, notably limited awareness and injection-related apprehension, which may impact implementation success. Beyond individual-level barriers such as fear of injections, successful rollout of LAI-ART in LMICs will require strengthened health infrastructure, trained personnel, and reliable supply chains to ensure consistent access and delivery of injectables [9]. In general, CAB + RPV LA maintained high levels of satisfaction and acceptance [37]. The higher rate of interest expressed in our study is likely due to the enormous challenges faced by APHI as compared to adults [38]. In fact, interest to LAI-ART among APHI was largely driven by the experience of pill fatigue and the hope that this long-acting regimen could greatly improve their adherence to treatment (Fig 2). This is in line with studies showing that LAI-ART could be crucial for those struggling with the daily reminder of their HIV status and the stigma associated with it. [39]. The burden of daily medication can lead to feelings of isolation and anxiety, henceforth underscoring

the promise of LAI-ART to maintain VS, even among patient on daily pill experiencing virological failure [40]. An observational cohort assessing retention in care among adolescents and youths living with HIV on LAI CAB + RPV reported 100% engagement in care and VS during the study follow-up [11].

Long-acting CAB + RPV is not yet available in most LMICs. However, youth-led advocacy is ongoing, and in 2023, youths published the Lusaka declaration in which they call to action to make the LAI-ART available for them. In fact, they believe that making LAI-ART availability will likely offer more choice for both treatment and prevention and improve the quality of their lives. The high interest to LAI-ART reported in the present study is in line with these global enthusiasms of adolescents and youths living with HIV about LAI-ART. However, fear of injections could impede uptake, since adolescents with a high fear of injections significantly less likely to express interest in LA-ART (Table 2). Similar data was observed in previous studies which equally highlighted the fear of needle/injection as a factor limiting adolescents' and youths' preference for LAI-ARVs [18,19]. Although only a minority (about 10%) of adolescents reported high fear of injections/needles, this remains a relevant implementation concern. As fear of injection was assessed using a single-item, non-validated measure, it should be interpreted as a pragmatic indicator of perceived concern. Still, as suggested in the literature, adolescent-friendly counselling, clear education on the injection process, peer and caregiver support, and youth-friendly delivery models may help reduce anxiety and improve acceptability during LAI-ART rollout.

This suggestion can be further justified by the fact that APHI without any knowledge of long acting were significantly less likely to be interested in LAI-ART. Our results align with a previous study revealing that interest in LAI-ART was significantly associated with prior knowledge [19]. Beyond the mere dissemination of information, achieving optimal success in the introduction of LAI-ART in LMICs requires active youth engagement in the design, implementation, monitoring, and evaluation of new therapeutic strategies aimed at improving their wellbeing. Complementary measures such as community-based education, nurse-led LAI-ART delivery, and integration into adolescent-friendly services can further support this process by addressing misconceptions early and mitigating common implementation barriers.

We showed in this study that apart from fear of injection and the lack of knowledge which were both negative predictors for the interest to LAI-ART, APHI express high interest to LAI-ART regardless of sex, educational level, residential areas, adherence levels, viral load levels or other therapeutic histories. Moreover, even though the current rollout of the MDSD models shows encouraging results, if LAI-ART were made available, most adolescents (77.1%) would prefer LAI-ART over MDSD.

Adherence in this study was self-reported and may be subject to recall bias; however, this limitation does not impact the primary outcome, which focused on interest in LAI-ART. Additionally, the cross-sectional design and recruitment of participants from clinics within a single region may introduce selection bias and limit the generalizability of our findings, particularly for adolescents in other regions who may have different demographic, socio-economic and clinical characteristics. This design also captures interest at a single time point; this interest, based on limited information, may change as adolescents gain a more comprehensive understanding of LAI-ART or as it becomes available in practice. These findings therefore underscore the need for longitudinal implementation studies to assess how initial preferences translate into actual uptake once LAI-ART becomes available in routine care.

However, our findings address a critical evidence gap on long-acting ART among adolescents in LMICs. They support the goals of the CIPHER-ADOLA study by generating context-specific evidence to inform the introduction of LAI-ART as an innovative treatment option with the potential to overcome persistent challenges faced by APHI in sub-Saharan Africa.

In conclusion, in this population of Cameroonian APHI among whom one-quarter have poor adherence, knowledge or awareness on LAI-ART remains low, as regimens are yet to be locally available. Upon awareness, the greatest majority expresses interest in receiving LAI-ART as a mean to mitigate adherence and to relieve pill burden. Nonetheless, limited knowledge/awareness on LAI-ART as well as fear of injections represent a barrier for uptake. Thus, community interventions for awareness and adherence to injectables are key interventions for a preparatory phase toward a successful introduction and scale-up of LAI-ART among APHI living within similar low- and middle- income countries.

# Supporting information

**S1 Data. Data underlying the results.**
(XLSX)

# Acknowledgments

Our findings reported in this paper are dedicated to late Gouissi Hyacinthe (PhD research fellow under the CIPHER-ADOLA study) who immensely contributed to the implementation of this study but tragically passed away in March 2025. We also acknowledge the massive contribution of the community of young people living with HIV in Cameroon, particularly the members of the network of positive young Cameroonians, namely *"RECAJ+" (Réseau des Jeunes Camerounais Positifs)*.

# Author contributions

**Conceptualization:** YAGAI BOUBA, Aude Christelle Ka'e, Vittorio Colizzi, Francesca Ceccherini-Silberstein, Carlo-Federico Perno, Alexis Ndjolo, Maria Mercedes Santoro, Joseph Fokam.

**Data curation:** YAGAI BOUBA, Aude Christelle Ka'e, Cynthia Ayafor, Lum Forgwei, Jeremiah Efakika Gabisa, Nadine Nguendjoung Fainguem, Michel Carlos Tommo Tchouaket, Desire Takou, Félicité Noukayo, Ezechiel Ngoufack Jagni Semengue, Naomi-Karell Etame.

**Formal analysis:** YAGAI BOUBA, Aude Christelle Ka'e, Cynthia Ayafor, Jeremiah Efakika Gabisa, Alex Durand Nka, Daniele Armenia, Joseph Fokam.

**Funding acquisition:** YAGAI BOUBA, Francesca Ceccherini-Silberstein, Carlo-Federico Perno, Alexis Ndjolo, Maria Mercedes Santoro, Joseph Fokam.

**Investigation:** YAGAI BOUBA, Aude Christelle Ka'e, Cynthia Ayafor, Lum Forgwei, Jeremiah Efakika Gabisa, Daniel Mabongo, Alex Durand Nka, Rita Ekwoge Mejane, Suzie Ndiang Tetang, Rachel Simo Kamgaing, Francis Ndongo Ateba, Nadine Nguendjoung Fainguem, Michel Carlos Tommo Tchouaket, Desire Takou, Nelly Kamgaing, Michelle Aloum Menye, Ezechiel Ngoufack Jagni Semengue, Roland Wome Basseck, Agabus Wiadamong, Abdou Rahamani Gnambi, Catherine Eyenga, Naomi-Karell Etame, Aurelie Minelle Kengni Ngueko, Larissa Gaëlle Moko Fotso, Junie Flore Yimga, Grace Anong Beloumou, Collins Ambe Chenwi, Alice Ketchaji, Hyppolite Kuekou Tchidjou, Rogers Awoh Ajeh.

**Methodology:** YAGAI BOUBA, Aude Christelle Ka'e, Cynthia Ayafor, Jeremiah Efakika Gabisa, Alex Durand Nka, Suzie Ndiang Tetang, Rachel Simo Kamgaing, Francis Ndongo Ateba, Nadine Nguendjoung Fainguem, Michel Carlos Tommo Tchouaket, Nelly Kamgaing, Ezechiel Ngoufack Jagni Semengue, Samuel Martin Sosso, Francesca Ceccherini-Silberstein, Carlo-Federico Perno, Maria Mercedes Santoro, Joseph Fokam.

**Project administration:** YAGAI BOUBA, Paul Ndombo Koki, Francesca Ceccherini-Silberstein, Alexis Ndjolo, Maria Mercedes Santoro, Joseph Fokam.

**Supervision:** Gregory Edie Halle Ekane, Paul Ndombo Koki, Daniele Armenia, Vittorio Colizzi, Gianluca Russo, Stefano D'amelio, Francesca Ceccherini-Silberstein, Carlo-Federico Perno, Alexis Ndjolo, Maria Mercedes Santoro, Joseph Fokam.

**Visualization:** Aude Christelle Ka'e, Cynthia Ayafor, Jeremiah Efakika Gabisa, Nelly Kamgaing, Hyppolite Kuekou Tchidjou, Gregory Edie Halle Ekane, Daniele Armenia, Francesca Ceccherini-Silberstein, Carlo-Federico Perno, Maria Mercedes Santoro, Joseph Fokam.

**Writing – original draft:** YAGAI BOUBA, Aude Christelle Ka'e, Cynthia Ayafor, Lum Forgwei, Jeremiah Efakika Gabisa, Daniel Mabongo, Alex Durand Nka, Rita Ekwoge Mejane, Suzie Ndiang Tetang, Rachel Simo Kamgaing, Francis

Ndongo Ateba, Nadine Nguendjoung Fainguem, Michel Carlos Tommo Tchouaket, Desire Takou, Félicité Noukayo, Ezechiel Ngoufack Jagni Semengue, Roland Wome Basseck, Agabus Wiadamong, Abdou Rahamani Gnambi, Naomi-Karell Etame, Junie Flore Yimga, Grace Anong Beloumou, Collins Ambe Chenwi, Alice Ketchaji, Hyppolite Kuekou Tchidjou, Alexis Ndjolo.

**Writing – review & editing:** YAGAI BOUBA, Aude Christelle Ka'e, Samuel Martin Sosso, Gregory Edie Halle Ekane, Paul Ndombo Koki, Daniele Armenia, Vittorio Colizzi, Gianluca Russo, Stefano D'amelio, Francesca Ceccherini-Silberstein, Carlo-Federico Perno, Maria Mercedes Santoro, Joseph Fokam.

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
