## [Decision Letter · Decision Letter 0]

16 Mar 2026

PGPH-D-25-03641

Interest in long-acting injectable ART among adolescents with perinatally acquired HIV in Cameroon: implications for implementation in developing countries

Dear Dr.YAGAI BOUBA

Thank you for submitting your manuscript to PLOS Global Public Health. After careful consideration, we feel that it has merit but does not fully meet PLOS Global Public Health’s publication criteria as it currently stands. Therefore, we invite you to submit a revised version of the manuscript that addresses the points raised during the review process.

We look forward to receiving your revised manuscript.

Kind regards,

Henry Zakumumpa, PhD

Academic Editor

**Journal Requirements:**

1. In the online submission form, you indicated that “Data is available upon request from the corresponding author.”.

3. Uploaded as supplementary information.

**Additional Editor Comments (if provided):**

We are delighted to share comments from our reviewers regarding your submission.

Please provide a point by point response to the raised comments so we can move swiftly to a decision.

Endevour to concede the limitations of your study such as the sample size and resist the temptation to strongly state the findings.

Reviewers' comments:

Reviewer's Responses to Questions

**Comments to the Author**

1. Does this manuscript meet PLOS Global Public Health’s publication criteria? Is the manuscript technically sound, and do the data support the conclusions? The manuscript must describe methodologically and ethically rigorous research with conclusions that are appropriately drawn based on the data presented.

Reviewer #1: Yes

Reviewer #2: Yes

2. Has the statistical analysis been performed appropriately and rigorously?

Reviewer #1: Yes

Reviewer #2: Yes

3. Have the authors made all data underlying the findings in their manuscript fully available (please refer to the Data Availability Statement at the start of the manuscript PDF file)?

Reviewer #1: No

Reviewer #2: Yes

4. Is the manuscript presented in an intelligible fashion and written in standard English?

Reviewer #1: Yes

Reviewer #2: Yes

5. Review Comments to the Author

**Reviewer #1:** Manuscript ID: PGPH-D-25-03641

Title: Interest in long-acting injectable ART among adolescents with perinatally acquired HIV in Cameroon.

General Assessment

This manuscript addresses a timely and important gap by examining interest in long-acting injectable ART (LAI-ART) among adolescents with perinatally acquired HIV in Cameroon. The study is well designed, ethically conducted, and clearly written. The findings are highly relevant to the future rollout of LAI-ART in low- and middle-income countries. Overall, the manuscript is suitable for publication after minor to moderate revisions.

Major Comments

1. Interpretation of High Interest in LAI-ART

The reported level of interest (92.3%) is striking. As LAI-ART is not yet available locally, this likely reflects hypothetical acceptability. The authors should acknowledge this more explicitly and clarify what information (e.g., injection frequency, clinic visits, potential side effects) participants received before expressing interest.

2. Measurement of “Interest” and “Knowledge.”

The operational definitions of interest and knowledge are pragmatic but limited. Clarification is needed on whether “interest” reflects firm intention versus general preference. The discussion should reflect on how limited or superficial knowledge of LAI-ART may affect real-world uptake.

3. Fear of Injection as a Key Predictor

Fear of injection is an important and policy-relevant finding. The authors should briefly justify the use of a single-item, non-validated measure and discuss how injection fear might be addressed through counselling or adolescent-friendly education during rollout.

4. Generalizability

Most participants were recruited from urban clinics in one region. The limitations section should more clearly address potential selection bias and implications for adolescents in rural settings or those not engaged in care.

Minor Comments

1. Ensure consistency in viral suppression definitions (<50 vs <1000 copies/mL) and clarify their analytical use.

2. Table 1 is informative but dense; minor simplification would improve readability.

3. Figures would benefit from brief interpretive captions.

4. Minor typographical and formatting issues (e.g., spacing in “LAI-ART”) should be corrected.

5. PLOS generally encourages open data sharing; the authors may wish to briefly justify the “data available upon request” approach in light of ethical considerations involving

Ethics and publication considerations

Ethical approvals, assent and consent procedures, funding sources, and conflicts of interest are clearly reported and appropriate. I have no concerns regarding dual publication, research ethics, or publication ethics.

Recommendation

This is a strong and timely manuscript with clear implications for adolescent HIV care and future LAI-ART implementation in LMICs. However, minor to moderate revision is required. Addressing the points above will further strengthen its clarity and impact.

**Reviewer #2:** This is well written and important study that provides crucial, context specific evidence to inform the future rollout of LAI-ART for adolescents in Cameroon and similar settings. The manuscript is clear, the methodology is robust, and the findings are significant for implementation science.

Major comments:

1.Data Availability Statement: There is a discrepancy that must be resolved before publication In the manuscript draft(page 3). the statement reads" Data is available upon request from the corresponding author. " However, the PLOS data Policy requires authors to make all data underlying the findings fully available without restriction. Please revise this statement to either deposit the data in a public repository or provide it within the supporting Information files. An "available upon request " statement is generally not compliant with PLOS policy unless there are documented ethical or legal restrictions, which are not mentioned here.

Minor Comments/Suggestions:

1.Introduction (Lines 87-92): The sentence " Inconsistent adherence reflects limited access...all which erode the discipline required for daily dosing" is somewhat long and dense. Consider breaking it into two sentences for improved readability.

2. Methods(Line 17) Please clarify the formula notation for standard sample size calculations.

3.Results (Table 1) The footnote for a"" is missing from the table itself. Please add the corresponding symbol (e.g.,*) next to "Age at disclosure" in the table to match footnote.

4.Discussion(Line 324-325): The phrasing "nine in ten of Cameroonian APHI wanted LAI-ART" is slightly informal. Consider rephrasing to "Over 90% of Cameroonian APHI expressed interest in LAI-ART" for consistency with the abstract and results.

5.Discussion (Lines 391-393): The limitation paragraph is good, but it could be strengthened by briefly noting that cross sectional design captures interest at a single point in time, and that interest might change as adolescents gain more information or as LAI-ART becomes available. This also links well to your call for longitudinal Implementation research.

Conclusion:

The manuscript is technically sound, ethically rigorous, and makes a valuable contribution to the literature. Addressing the data availability statement and the minor comments above will strengthen the paper. I recommend Minor Revisions.

6. PLOS authors have the option to publish the peer review history of their article (what does this mean?). If published, this will include your full peer review and any attached files.

**Do you want your identity to be public for this peer review?** For information about this choice, including consent withdrawal, please see our Privacy Policy.

Reviewer #1: No

Reviewer #2: No

 Figure Resubmissions:

---

## [Decision Letter · Decision Letter 1]

28 Apr 2026

Interest in long-acting injectable ART among adolescents with perinatally acquired HIV in Cameroon: implications for implementation in developing countries

PGPH-D-25-03641R1

Dear YAGAI BOUBA,

We are pleased to inform you that your manuscript 'Interest in long-acting injectable ART among adolescents with perinatally acquired HIV in Cameroon: implications for implementation in developing countries' has been provisionally accepted for publication in PLOS Global Public Health.

Best regards,

Henry Zakumumpa, PhD

Academic Editor

Reviewer Comments (if any, and for reference):

Reviewer's Responses to Questions

**Comments to the Author**

1. If the authors have adequately addressed your comments raised in a previous round of review and you feel that this manuscript is now acceptable for publication, you may indicate that here to bypass the “Comments to the Author” section, enter your conflict of interest statement in the “Confidential to Editor” section, and submit your "Accept" recommendation.

Reviewer #1: All comments have been addressed

Reviewer #2: All comments have been addressed

2. Does this manuscript meet PLOS Global Public Health’s publication criteria? Is the manuscript technically sound, and do the data support the conclusions? The manuscript must describe methodologically and ethically rigorous research with conclusions that are appropriately drawn based on the data presented.

Reviewer #1: Yes

Reviewer #2: Yes

3. Has the statistical analysis been performed appropriately and rigorously?

Reviewer #1: Yes

Reviewer #2: Yes

4. Have the authors made all data underlying the findings in their manuscript fully available (please refer to the Data Availability Statement at the start of the manuscript PDF file)?

Reviewer #1: Yes

Reviewer #2: Yes

5. Is the manuscript presented in an intelligible fashion and written in standard English?

Reviewer #1: Yes

Reviewer #2: Yes

6. Review Comments to the Author

Reviewer #1: (No Response)

Reviewer #2: Thank you for the careful and thoughtful revision of your manuscript. I have reviewed your point-by-point responses and the revised manuscript with tracked changes. You have addressed the concerns raised during the first round of review thoroughly and respectfully. The manuscript has improved substantially in clarity, transparency, and scientific rigor.

Below, I offer my final assessment of how each major comment from the previous round has been addressed, along with a few observations. No further revisions are required, and I am pleased to recommend acceptance.

Response to Major comments from the first review

• Hypothetical acceptability of interest: You have appropriately clarified in the Methods section (lines 204) that interest was assessed after participants received brief information about an injectable regimen administered every eight weeks, and you now explicitly acknowledge in the Discussion (lines 401-409) that this represents hypothetical rather than actual willingness. This addressed the concern well.

• Definition of Interest and Knowledge: The revised manuscript now distinguishes more clearly between stated preference and firm intention, and the limitations section appropriately notes that interest may evolve as adolescents gain more comprehensive information about side effects, logistical requirements, and long-term implications. This is a balanced and honest representation.

• Fear of Injection Measurement: You have provided a reasonable justification for the single-item, non-validated measure as a pragmatic field-based indicator. More importantly, you have expanded the Discission (lines 377-383) to include concrete, actionable strategies to mitigate injection fear during rollout, including adolescent-friendly counselling, peer and caregiver support, and youth-friendly service delivery. This strengthens the policy relevance of your findings.

• Generalizability and selection bias: The limitations section (lines 401-409) no longer clearly acknowledges potential selection bias due to recruitment primarily from one region (Centre region) and the predominance of urban clinics, while noting that rural representation was limited to one district hospital. This is appropriately cautious and does not overclaim the findings.

• Consistency of viral suppression definitions: You have now clearly distinguished between viral suppression (<1000 copies/ML) for programmatic purposes and viral undetectability. (<50 copies/mL) for LAI/ART eligibility. This is applied consistently throughout the manuscript, resolving the earlier confusion.

• Table 1 readability: The revised table is indeed clearer and less dense. Removing the CD4 stratification from the table while retaining it in the text was a sensible decision.

• Figure Captions: The legends for Figures 1 and 2 now include brief interpretive sentences, which improve accessibility for readers.

• Data availability statement: You have resolved the prior discrepancy by providing a de-identified dataset and data dictionary as supporting information files. This now fully complies with PLOS data-sharing policies.

In conclusion, I am satisfied that the authors have responded fully and appropriately to all concerns raised in the first round of review. The manuscript will be of interest to HIV programme managers, Paediatric and adolescent HIV clinicians, and implementation scientists working in low and middle-income countries. No further peer review is required from my side.

7. PLOS authors have the option to publish the peer review history of their article (what does this mean?). If published, this will include your full peer review and any attached files.

**Do you want your identity to be public for this peer review?** For information about this choice, including consent withdrawal, please see our Privacy Policy.

Reviewer #1: No

Reviewer #2: **Yes:** JANE KALENGA NKATYA
